Prediction of wheat fusarium head blight severity levels in southern Henan based on K-means-SMOTE and XGBoost algorithms

Sun Xiaoyun 1
Su Shuaiming 1
Wang Qiang 1
Xiong Shufeng 1
Li Yanting 2
Peng Hong 3
Shi Lei 1 shilei@henau.edu.cn
1 College of Information and Management Science, Henan Agriculture University , Zhengzhou, Henan , China
2 College of Computer Science and Technology, Zhengzhou University of Light Industry , Zhengzhou, Henan , China
3 Plant Protection and Quarantine Station of Henan Province , Zhengzhou, Henan , China
Fasi Massimiliano
Electronic publication date: 2025 Mar 31
Publication date: 2025
Volume: 11
Electronic Location ID: e2638
Received 2024 Aug 23; Accepted 2024 Dec 6
Copyright: © 2025 Sun et al.
Copyright year: 2025
Copyright holder: Sun et al.
License: This is an open access article distributed under the terms of the Creative Commons Attribution License, which permits unrestricted use, distribution, reproduction and adaptation in any medium and for any purpose provided that it is properly attributed. For attribution, the original author(s), title, publication source (PeerJ Computer Science) and either DOI or URL of the article must be cited.
License URL: https://creativecommons.org/licenses/by/4.0/

Keywords: FHB severity prediction, Data imbalance, Feature selection, K-means-SMOTE, XGBoost

Funding: National Natural Science Foundation of China 31501225 and 62106233 Natural Science Foundation of Henan Province of China 232300420186 and 242301420143 Joint Fund of Science and Technology Research and Development Plan of Henan Province 222301420113 Science and Technology Research Project of Henan Province 242102211057 This work was supported by the National Natural Science Foundation of China (No. 31501225 and No. 62106233), the Natural Science Foundation of Henan Province of China (No. 232300420186 and No. 242301420143), Joint Fund of Science and Technology Research and Development Plan of Henan Province (No. 222301420113), and Science and Technology Research Project of Henan Province (No. 242102211057). The funders had no role in study design, data collection and analysis, decision to publish, or preparation of the manuscript.

==============================
Fusarium head blight (FHB) is a destructive disease which adversely affects the yield of wheat. The occurrence and epidemic of wheat FHB are closely related to meteorological information. Firstly, by analyzing eight meteorological factors—rainfall (RAIN), average sunshine hours (ASH), average wind speed (AWS), average temperature (AT), highest temperature (HT), lowest temperature (LT), average relative humidity (ARH), and maximum temperature difference (MTD)—specific periods closely related to wheat FHB severity are identified. Based on this, a dataset for wheat FHB severity is constructed. After that, the wheat FHB severity levels are divided into four levels, and actual field data shows that the proportion of data for the high prevalence severity level is relatively small. To address data imbalance, the K-means-synthetic minority over-sampling technique (K-means-SMOTE) method is introduced to increase samples of underrepresented severity levels. Subsequently, a wheat FHB severity prediction model based on K-means-SMOTE and extreme gradient boosting (XGBoost) is constructed. Lastly, by combining the rankings of meteorological factors provided by the model and the biological characteristics of wheat FHB, the number of meteorological factors is reduced from eight to four (AWS 4.24–4.28, RAIN 4.5–4.19, ARH 4.12–4.16, LT 4.19–4.23), the accuracy and recall of the model remained unchanged at 0.8936, the F1 score increased from 0.8851 to 0.8898, and the precision decreased from 0.9249 to 0.9058. Although the precision has slightly decreased, most of the other evaluation indicators of the model remain unchanged or have improved, therefore the model is considered effective. Finally, comparative experiments with eight other models demonstrate the superiority of this approach.

Introduction

In recent years, the occurrence of wheat fusarium head blight (FHB) in China has increased, posing a serious threat to wheat production and food safety (Wang, Li & Su, 2023). Among the 54 main wheat diseases and pests in China, FHB is one of the most severe (Huang, Jiang & Li, 2020). Wheat FHB can significantly reduce yield, lower grain quality, and even cause complete crop loss (Sun & Yin, 2022; Kang, 2021; Wang et al., 2022). Typically, wheat FHB reduces yield by 10% to 20%, but in severe cases, losses can reach 45%, seriously affecting harvests (Li, Liu & Shen, 2023). The disease can occur at all growth stages, especially during heading and flowering, continuous rainy weather further increases the risk (Xu, 2022; Pang, 2023). Global warming and straw return have made the occurrence of wheat FHB more frequent and unpredictable, complicating its management. Traditional monitoring strategies are no longer sufficient for effective control. Since wheat FHB occurrence is closely linked to meteorological factors (Zhao, 2024), identifying key meteorological factors and developing precise prediction models for wheat FHB severity is crucial for early warning and management.

Wheat FHB is a climate-driven disease closely linked to weather conditions (Chen, Kistler & Ma, 2019). Its yearly severity depends largely on changes in factors like rainfall, humidity, and temperature (Li et al., 2021, 2022; Matengu et al., 2024). Birr, Verreet & Klink (2019) used a regression model to predict FHB by analyzing the impact of weather on the levels of Deoxynivalenol (DON) and Zearalenone (ZEA) toxins in wheat and corn. David, Marr & Schmale (2016) found that spore release from Fusarium graminearum increased at 15 °C and with higher humidity, moving farther at 25 °C under maximum humidity. Xu et al. (2016) identified key microclimatic conditions for FHB, showing that when air temperature is above 13 °C and relative humidity is over 70%, FHB is more likely to occur. They used weather data from stations to further refine this model, with key thresholds of 14.3 °C and 64.7% humidity, achieving accuracy rates between 77.8% and 80.0% in field tests from 2003, 2004, and 2008. The above research indicates that the occurrence of wheat FHB is closely related to meteorological factors.

In recent years, the rapid development of machine learning technology has attracted significant attention to the application of machine learning algorithms in predicting the severity of wheat FHB. As a powerful data analysis tool, machine learning can automatically extract patterns from large-scale data (Yang & Dong, 2024). Given the close relationship between the occurrence of wheat FHB and meteorological information, using machine learning models can effectively process meteorological data, enhancing the accuracy and efficiency of predictions. For example, Tao et al. (2021) analyzed the incidence of wheat FHB and meteorological information during key growth stages in 21 major wheat-producing counties (cities) in the Haihe Plain from 2001 to 2016, using stepwise regression analysis to select key meteorological factors, and developed a prediction framework combining multiple linear regression and boosted regression tree (BRT) models. Xiao et al. (2022) developed a disease prediction and progression simulation model based on mathematical models and remote sensing data, specifically using the susceptible-exposed-infected-recovered (SEIR) model to simulate the occurrence and progression of wheat FHB, combined with weather data and multi-source remote sensing data to improve prediction accuracy. Yu et al. (2023) established a wheat FHB prediction support vector regression (PSO-SVR) model optimized by particle swarm algorithm based on the data of wheat head blight incidence rates in Chuzhou City from 2005 to 2020 and the corresponding meteorological data, achieving a minimum mean squared error (MSE) of 0.9. The aforementioned research indicates that machine learning methods offer relatively simple predictive models for multiple meteorological factors with low learning costs. These studies effectively utilize machine learning techniques to process meteorological data, significantly enhancing the accuracy and efficiency of predicting the severity of wheat FHB. However, these studies did not take into account the issue of data imbalance that may arise with different severity levels in the prediction of wheat FHB.

Specifically, although there is a large volume of meteorological data, there is typically only one set of wheat FHB severity level data each year. Consequently, due to the wheat FHB data often shows significant imbalance due to low prevalence, posing challenges for statistically-based disease prediction models. This imbalance not only affects model training efficiency but also may lead to insufficient prediction of high-risk disease levels. However, few studies have effectively addressed the challenges associated with the uneven grading of wheat FHB severity levels. In order to solve the problem of data imbalance based on the characteristics of the data in this study, oversampling methods are needed. Chawla et al. (2002) proposed the synthetic minority over sampling technique (SMOTE) oversampling method in 2002. Based on this method, many variants have emerged, He et al. (2008) proposed adaptive synthetic sampling (ADASYN) in 2008 and Li et al. (2024) proposed hybrid density-based adaptive weighted corroborative representation (HDAWCR) in 2024. Compared to these methods, Douzas, Bacao & Last (2018) proposed the K-means-SMOTE in 2018, K-means-SMOTE combines K-means clustering and SMOTE oversampling by first clustering minority class samples and then generating new minority samples within each cluster, thus creating more representative samples, effectively addressing data imbalance, improving classification model performance, and reducing the risk of overfitting, the main feature of K-means-SMOTE lies in its combination of clustering and oversampling. But SMOTE and other methods directly generates new samples between minority class samples, potentially ignoring the internal structure of the data, resulting in new samples that are less representative than those generated by K-means-SMOTE. For example, Zeng et al. (2024) uses the K-means-SMOTE method to cluster minority class samples and then oversample them to generate more representative samples, solving the problem of imbalanced volunteer rating data and improving the performance of subsequent machine learning models.

After addressing data imbalance, selecting the appropriate machine learning algorithm is crucial for improving prediction accuracy. Compared to traditional machine learning models such as random forest (RF) (Breiman, 2001), gradient boosting decision trees (GBDT) (Friedman, 2001), and K-nearest neighbors (KNN) (Kramer & Kramer, 2013), extreme gradient boosting (XGBoost), introduced by Chen & Guestrin (2016) in 2016, offers unparalleled advantages in generalization performance, computational speed, and prediction accuracy (Liu et al., 2017; Mahmud et al., 2019; Nawar & Mouazen, 2019). XGBoost can handle unequal class distributions by adjusting sample weights, making it particularly suitable for predicting wheat FHB severity levels, where data for high severity levels are relatively scarce. As a gradient boosting model based on decision trees, XGBoost combines the strengths of both gradient boosting and decision tree models, offering superior efficiency and accuracy (Chen & Guestrin, 2016; Guo et al., 2022). Its core approach is to iteratively train decision trees, improving the model’s performance by optimizing the objective function with each iteration.

This study plans to address the following key issues: (1) effectively handle the imbalance in wheat FHB severity levels data, especially under high prevalence level data scarcity; (2) extract key features beneficial for disease severity prediction from a large amount of meteorological and crop health data; (3) construct an efficient and accurate prediction model for real-time monitoring and early warning of wheat FHB severity levels. To achieve these goals, this study employs a K-means-SMOTE-XGBoost approach, which integrates data balancing, feature selection, and predictive modeling to address the challenges of data imbalance and feature extraction. Through this research, the article aims to provide more scientific and precise decision support for wheat FHB prevention.

Materials and Methods

Collection of disease data and meteorological information

This study has collected basic data on the occurrence of wheat FHB in 17 cities in the southern part of Henan Province from 2016 to 2023, as shown in Fig. 1.

Figure 1 Map of the study area of 17 cities in the southern part of Henan Province.

The wheat FHB incidence data used in this study was obtained from the Plant Protection and Quarantine Station of Henan Province, while meteorological data was sourced from the Henan Meteorological Bureau (http://ha.cma.gov.cn/). The incidence data includes the annual average diseased ear rates for wheat from 2016 to 2023 across 17 cities. The meteorological dataset encompasses seven key factors—rainfall (RAIN), average sunshine hours (ASH), average wind speed (AWS), average temperature (AT), highest temperature (HT), lowest temperature (LT), and average relative humidity (ARH)—collected from January 1 to June 1 each year across 17 cities over the same period, capturing the entire growth cycle of wheat. Based on the investigation results and according to the national technical specification for the monitoring and reporting of wheat FHB, recommended national standard (GB/T) 15796-2011, wheat FHB is classified into four levels: not occurring (Level 0, spike infection rate <0.1%, no significant impact on wheat production), mild prevalence (Level 1, 0.1% ≤ spike infection rate <5%, causing partial yield reduction in wheat production), moderate prevalence (Level 2, 5% ≤ spike infection rate <10%, causing partial yield reduction in wheat production), and severe prevalence (Level 3, spike infection rate ≥10%, causing significant yield reduction in wheat production). In addition to data on spike infection rates, this study also collects and records meteorological information for the corresponding periods, including seven key meteorological variables: daily precipitation, wind speed, average temperature, maximum temperature, minimum temperature, relative humidity, and hours of sunshine from January 1 to June 1 each year from 2016 to 2023.

K-means-SMOTE

Due to the real-world field data on wheat FHB severity levels, the issue of sample imbalance is inevitable. Without proper handling, the prediction results for the wheat FHB severity levels would be biased towards the categories with larger sample sizes, leading to larger prediction errors for categories with smaller sample sizes. In the current collected data, there are 64 instances of wheat FHB at severity Level 0, 58 instances at severity Level 1, six instances at severity Level 2, and eight instances at severity Level 3. Clearly, there is a significant difference in the number of data points for each level, with an imbalance ratio (IR) greater than 10:1 (Jing et al., 2021). For such highly imbalanced datasets, classification results often show high accuracy for the majority classes while approaching zero for the minority classes. This can lead to a decrease in the overall accuracy of the model. To address the issue of data imbalance, this study uses the K-means-SMOTE oversampling technique. According to literature (Douzas, Bacao & Last, 2018) due to the smaller number of instances in certain classes, typical classifiers tend to overlook the detection of minority classes.

SMOTE is one of the oversampling techniques used to solve this problem (Chawla et al., 2002). It generates minority instances within overlapping regions. However, SMOTE randomly synthesizes minority instances along the line connecting minority instances and their selected nearest neighbors, ignoring nearby majority instances. In contrast, the K-means-SMOTE technique applies the K-means clustering algorithm to group minority instances in the dataset and then performs oversampling within each cluster. This approach avoids the problem of SMOTE ignoring nearby majority instances. Within each cluster, the K-means-SMOTE technique randomly synthesizes minority instances along the line connecting minority instances and their selected nearest neighbors, thus maintaining the distribution characteristics of the data. This method better captures the local structure of minority class samples, thereby improving model accuracy performance. The basic idea is to balance the dataset by synthesizing new minority class samples. The algorithm of the K-means-SMOTE technique is described in Douzas, Bacao & Last (2018), with the specific steps shown in Table 1.

Table 1 K-means-SMOTE algorithm.

Algorithm 1 Proposed method based on K-means and SMOTE.	
Require: X (matrix of observations)	
         y (target vector)	
         n (number of samples to be generated)	
         k (number of clusters to be found by K-means)	
         irt (imbalance ratio threshold)	
         knn (number of nearest neighbors considered by SMOTE)	
         de (exponent used for computation of density; defaults to the number of features in X)	
1: begin	
2: // Step 1: Cluster the input space and filter clusters with more minority instances than majority instances.	
3: clusters←K-means(X)	
4: filtered clusters←∅	
5: for c∈clusters do	
6:    imbalance ratio←majority count(c)+1minority count(c)+1	
7:    if imbalance ratio<irt then	
8:       filtered clusters←filtered clusters∪{c}	
9:   end if	
10: end for	
11: // Step 2: For each filtered cluster, compute the sampling weight based on its minority density.	
12: for f∈filtered clusters do	
13:     average minority distance(f)←mean(euclidean distances(f))	
14:     density factor(f)←minority count(f)average minority distance(f)de	
15:    sparsity factor(f)←1density factor(f)	
16: end for	
17: sparsity sum←∑f∈filtered clusterssparsity factor(f)	
18: sampling weight(f)←sparsity factor(f)sparsity sum	
19: // Step 3: Oversample each filtered cluster using SMOTE. The number of samples to be generated is computed using the sampling weight.	
20: generated samples←∅	
21: for f∈filtered clusters do	
22:   number of samples←||n×sampling weight(f)||	
23:   generated samples←generated samples∪SMOTE(f,number of samples,knn)	
24: end for	
25: return generated samples	
26: end	

Computational complexity is an important indicator to measure the efficiency of an algorithm. The computational complexity of K-means-SMOTE primarily lies in the clustering step, with a time complexity of O(n⋅k⋅t), where n is the number of samples, k is the number of clusters and t is the number of iterations of the K-means algorithm. Although the computational cost may increase significantly with larger data sizes, this complexity can be effectively controlled through reasonable selection of the number of clusters k, and optimization of the algorithm.

In contrast, the traditional SMOTE and ADASYN methods have a computational complexity of O(n⋅k⋅log⁡n) mainly arising from the need to find the k nearest neighbors for each minority class sample. When dealing with larger sample sizes, SMOTE and ADASYN may face a higher computational burden, especially when processing datasets with high-dimensional features, as the complexity of k-nearest neighbor searches further increases.

Overall, the efficiency of the K-means-SMOTE method is more guaranteed compared to other methods.

XGBoost

In each iteration, XGBoost builds a new decision tree model based on the residuals between the previous model’s predictions and the actual labels. The prediction results of the new model are then weighted and added to the previous model results to obtain the final integrated prediction result. The XGBoost prediction model can be expressed as

(1) y^i(t)=∑k=1tfk(xi),fk∈F.

In the formula: k represents the number of decision trees, F corresponds to the set of all decision trees, fk is the k-th decision tree generated in the k-th iteration.

The loss function of the results can be represented by the predicted value yi and the actual value ŷi.

(2) L=∑i=1nl(yi,y^i).

The objective function O consists of the model’s loss function L and a regularization term Ω that penalizes model complexity.

(3) O=∑i=1nl(yi,y^i)+∑i=1nΩ(fi).

Finally, according to the additive model, an overall XGBoost prediction model is obtained, as shown in Fig. 2.

Figure 2 XGBoost algorithmic framework diagram.

In the figure: f1…fk—Decision tree outputs predicted values; Number of k decision trees.

Similarly, this study selects XGBoost as the model algorithm mainly due to its significant advantages in computational efficiency and optimization performance. The computational complexity of XGBoost is O(n⋅m⋅log⁡m), where n represents the number of samples and m represents the number of features. By constructing and optimizing multiple decision trees through incremental learning based on weighted loss, XGBoost significantly improves model training efficiency. In contrast, the computational complexity of GBDT is also O(n⋅m⋅log⁡m), but GBDT constructs trees sequentially and lacks parallel processing, making XGBoost generally faster in training, especially on large datasets.

The computational complexity of RF is O(t⋅n⋅log⁡m), where t is the number of trees, n is the number of samples, and m is the number of features. RF predicts by simultaneously constructing multiple decision trees, and its computational complexity increases linearly with the number of trees. Although RF has strong parallel capabilities, the construction of each tree is independent, lacking the mechanism in XGBoost that optimizes each tree through weighted loss. Therefore, XGBoost demonstrates better performance when handling large datasets, effectively reducing training time and preventing overfitting.

Construction of wheat fhb severity levels monitoring model

Model construction

Based on the principles of K-means-SMOTE and the XGBoost algorithm, the steps to establish the wheat FHB severity levels monitoring model are as follows:

1. Data Division: Based on historical data and previous research, select meteorological factors closely related to the occurrence and development of wheat FHB severity levels, and divide the data to prepare for correlation analysis.

2. Correlation Analysis: Analyze the correlation between various meteorological factors and wheat FHB severity levels using the Kendall coefficient, identify key periods that have a significant impact on the prediction of wheat FHB severity levels, and organize the corresponding dataset.

3. K-means-SMOTE Oversampling: To address the class imbalance issue in the dataset, the K-means-SMOTE algorithm is used to enhance minority class samples.

4. XGBoost Classification Prediction and Meteorological Factor Selection: Further train the model using the XGBoost model, which combines the predictive capabilities of multiple decision trees and optimizes model performance through the gradient boosting algorithm. This is suitable for handling large-scale complex data. The XGBoost model also provides a ranking of the importance of meteorological factors, allowing for the selection of more strongly correlated meteorological factors.

5. Reconstruct the Dataset and Retrain the Model Using XGBoost: Use the data of the four selected key meteorological factors as the new dataset for model training, continuously adjusting model parameters until the model achieves optimal prediction performance.

The flowchart is shown in Fig. 3.

Figure 3 Flow chart of the construction of wheat FHB severity levels model.

Data division

This study processes the seven meteorological variables for each county (city, district), considering the high incidence period of wheat FHB to be around the heading and flowering stages (Zhou, 2024). For southern Henan, the period from March 1 to May 20 is considered. To accurately identify the specific stage at which each meteorological variable is most significantly correlated with wheat FHB severity levels, this study integrates all relevant data from 17 cities from 2016 to 2023, dividing the data into a total of 81 time periods.

Specifically, precipitation (RAIN) is calculated on a 15-day cycle, from March 1 to May 20 each year, with the total rainfall calculated every 15 days. For example, labels such as RAIN3.1−3.15, RAIN3.2−3.16, etc., up to RAIN5.6−5.20, represent the total rainfall from March 1 to March 15, March 2 to March 16, and so on, up to May 6 to May 20.

The other seven meteorological factors are divided into 5-day periods. The total sunshine hours (ASH) for each 5-day period are calculated from March 1 to May 20 each year. The average wind speed (AWS), average temperature (AT), highest temperature (HT), lowest temperature (LT), average relative humidity (ARH), and the maximum temperature difference (MTD) between the highest and lowest temperatures within each 5-day period are also calculated. For example, labels such as ASH3.1−3.5, ASH3.2−3.6 , etc., up to ASH5.16−5.20, represent the total sunshine hours from March 1 to March 5, March 2 to March 6, and so on, up to May 16 to May 20.

Then, the meteorological data for the same period from all counties are combined with the corresponding wheat FHB severity level data into one dataset. Thus, for each group of meteorological data (e.g., the group labeled 3.1–3.5), the corresponding data from 17 cities from 2016 to 2023 are aggregated to form the first dataset. This process is repeated, forming a total of 81 such datasets, each containing the meteorological information and wheat FHB severity levels data for all relevant counties within a specific time period. Before feature selection, we first conducted a statistical analysis of missing values and outliers to ensure the integrity and accuracy of the data.

Feature selection

The Kendall’s tau coefficient is a non-parametric statistical method used to measure the ordinal association between two random variables, with values ranging between −1 and 1; It is based on the number of concordant and discordant pairs in the data, providing a way to measure the strength and direction of the monotonic relationship between two variables (Bae & Kim, 2014). In practical applications, the Kendall’s tau coefficient does not require assumptions about the data distribution, has good adaptability to small samples and non-normally distributed data, and is relatively robust to outliers. The Kendall’s tau coefficient algorithm is as follows:

Suppose there are two sequences, X and Y, each with elements. The i-th element ( 1≤i≤N) in the two sequences is represented by Xi and Yi respectively. When any two elements ( Xi, Yi) and ( Xj, Yj) in the sets and have the same order, i.e., Xi>Xj and Yi>Yj or Xi<Xj and Yi<Yj, these two elements are considered concordant. When Xi<Xj and Yi>Yj or Xi>Xj and Yi<Yj occur, these two elements are considered discordant. When Xi=Xj or Yi=Yj occur, these two elements are neither concordant nor discordant. Then the calculation of Kendall’s tau correlation coefficient is as follows:

(4) τ=C−D(N0−N1)(N0−N2),

where C represents the number of concordant pairs, D represents the number of discordant pairs, and N represents the total number of elements.

(5) N0=12N(N−1),

(6) N1=∑i=1s112Ui(Ui−1),

(7) N2=∑i=1s212Vi(Vi−1),

where S1 and S2 represent the number of types of repeated elements in X and Y respectively, and Ui and Vi represent the number of elements in the i-th set of identical elements in X and Y, respectively.

Using Kendall’s tau coefficient, we analyzed the correlation between different meteorological factors during specific periods and the wheat FHB severity levels. The specific approach involved applying Kendall’s tau coefficient to the 81 pre-split datasets sequentially. For each meteorological factor, we selected the period with the highest correlation, and so forth. This resulted in obtaining the data for the eight meteorological factors with the strongest correlations, along with the corresponding year’s wheat FHB severity data, thus forming a new dataset. Eight key meteorological factors that influence the occurrence of wheat FHB severity levels were identified: total precipitation RAIN4.5−4.19, average wind speed AWS4.24−4.28, average temperature AT3.7−3.11, highest temperature HT3.7−3.11, lowest temperature LT4.19−4.23, average relative humidity ARH4.12−4.16, average sunshine hours ASH3.4−3.8, and maximum temperature difference MTD5.14−5.18.

To visually display the correlation between each selected meteorological factor and the wheat FHB severity levels, we calculated the correlations using Kendall’s tau coefficient and presented the results in the form of a heatmap, as shown in Fig. 4. The absolute value of the correlation between each meteorological factor and the wheat FHB severity levels exceeded 30%, which confirms the significant role of key meteorological factors in the development of FHB. This indicates that RAIN, LT, and ARH are positively correlated with the levels of FHB, which aligns with Zhou’s (2024) explanation of the disease’s development patterns. This provides a scientific basis for further formulating targeted prevention and control measures.

Figure 4 Kendall correlation coefficient heatmap of eight meteorological factors with wheat FHB severity levels.

Results

Parameter selection

Parameter selection is a crucial part of the experiment, based on the oversamplers listed in Fig. 2 as benchmarks for evaluating the proposed method, as well as the hyperparameter sets used for each oversampler. This work aims to create comparability between oversamplers; therefore, it is most important for oversamplers to achieve the same IR. Since there are only six level 2 data in the dataset, K (number of clusters to be found by K-means) in K-means-SMOTE is set to 1, 2, if K is greater than or equal to 3, there may be only two data points in each cluster, which would prevent effective oversampling. In comparison to other oversampling methods such as SMOTE and ADASYN, KNN (number of nearest neighbors considered by SMOTE) is set to 1, 2, 3, 4. In order to make IR as close to 1:1 as possible, the n2 (the number of samples to be generated at level 2) of all oversampling methods is set to be 8, 9, and 10 times that of level 2 samples, and the n3 (the number of samples to be generated at level 3) is set to be six, seven, and eight times that of level 3 samples. The dataset was randomly divided into 80% training set and 20% test set to evaluate the model’s performance under different parameter settings. The specific parameter settings are shown in the Table 2 below.

Table 2 Parameter settings for different oversampling methods.

Parameter	Explain	Parameter pool	
K	Number of clusters to be found by K-means	[1, 2]	
Knn	Number of nearest neighbors considered by SMOTE	[1, 2, 3, 4]	
n2	The number of samples to be generated at level 2	[8, 9, 10]	
n3	The number of samples to be generated at level 3	[6, 7, 8]	

To further optimize the model’s performance, this study tuned the key parameters of the XGBoost model. Specific settings are shown in Table 3. During the XGBoost model parameter tuning process, the GridSearchCV method was used to find the optimal parameter combination. Specifically, grid search traverses each parameter combination and evaluates the performance of each combination through cross-validation to find the best parameter settings.

Table 3 Parameter settings for different classification methods.

Description	Parameter pool	Parameter value	
learning_rate	[0.001, 0.01, 0.1, 1]	0.001	
max_depth	[3, 5, 7, 9]	3	
n_estimators	[10, 100, 200, 300]	100	

To demonstrate the reasonableness of the parameter pool and the chosen parameters, this research conducted a parameter sensitivity analysis.

First, n_estimators was fixed at 100 and the learning_rate at 0.001, systematically evaluating max_depth from 1 to 300 with a step size of 1. The results indicated that the model achieved its highest accuracy when max_depth was set to 3, highlighting optimal performance at this depth. As max_depth increased, accuracy remained stable at four and five but began to decline after reaching six and then remained unchanged, as shown in Fig. 5A.

Figure 5 The variation of model accuracy under different (A) max_depth, (B) n_estimators, and (C) learning_rates.

Afterwards, in the case of fixing max_depth at 3 and learning_rate at 0.001, the accuracy of n_estimators was evaluated from 1 to 300, which showed that the optimal solution was achieved when n_estimators was set to 13, and maintained it thereafter, as shown in Fig. 5B.

Finally, the parameters were fixed at max_depth = 3 and n_estimators = 100, and the learning_rate was varied from 0.001 to 1 in increments of 0.001. The results showed that the model reached its optimal value at the beginning and did not show any obvious pattern afterwards, as shown in Fig. 5C.

Then combine each oversampling method with each classifier for hyperparameter selection. Specifically, to achieve an IR as close to 1:1 as possible, the values for n2 and n3 are divided into three groups: n2 = 8 with n3 = 6, n2 = 9 with n3 = 7, and n2 = 10 with n3 = 8. For K-means-SMOTE, these three groups are combined with K = 1 and K = 2. For SMOTE and ADASYN, these three groups are combined with KNN = 1, KNN = 2, KNN = 3, and KNN = 4. It is important to note that the optimal results varied across different combinations. Subsequently, the parameter combinations for each oversampling method were tested with the classification models, and the combination that demonstrated comparatively better performance was ultimately selected, as shown in Table 4.

Table 4 Final parameter setting.

Parameter	Parameter value	
K	2	
KNN	1	
n2	8	
n3	6	
learning_rate	0.001	
max_depth	3	
n_estimators	100	

The selection of parameters is also in line with the expectations of this study, such as the final determination of KNN, as Sun et al. (2024) mentioned, blindly continuing to increase the k nearest neighbors value will undoubtedly increase the computational load, which is often undesirable while maintaining optimal accuracy. Therefore, based on these findings and experimental comparison results, this study chooses to fix the KNN value at 1.

At this point, the evaluation indicators of the model are shown in Table 5, with an accuracy of 0.8936, precision of 0.9249, recall of 0.8936, and F1 score of 0.8851. Additionally, the importance ranking of each meteorological factor is provided, in order: AWS4.24−4.28, RAIN4.5−4.19, MTD5.14−5.18, ARH4.12−4.16, HT3.7−3.11, LT4.19−4.23, ASH3.4−3.8 and AT3.7−3.11. The importance of each factor is 0.307662, 0.287486, 0.191471, 0.118757, 0.036119, 0.031564, 0.013735, and 0.013207, respectively.

Table 5 Results of different model combinations.

The bolded entries represent the best-performing results for the corresponding metrics among all the compared models.

Method	Accuracy	Precision	Recall	F1 score	
K-means-SMOTE-GBDT	0.8511	0.9059	0.8511	0.8482	
K-means-SMOTE-RF	0.8298	0.8364	0.8298	0.8199	
K-means-SMOTE-XGBoost	0.8936	0.9249	0.8936	0.8851	
K-means-SMOTE-KNN	0.8511	0.8797	0.8511	0.8190	
SMOTE-GBDT	0.8723	0.9149	0.8723	0.8578	
SMOTE-RF	0.8723	0.8704	0.8723	0.8605	
SMOTE-XGBoost	0.8723	0.9051	0.8723	0.8545	
SMOTE-KNN	0.8723	0.8971	0.8723	0.8518	
ADASYN-GBDT	0.7872	0.7890	0.7872	0.7728	
ADASYN-RF	0.7660	0.7577	0.7660	0.7473	
ADASYN-XGBoost	0.7660	0.7638	0.7660	0.7508	
ADASYN-KNN	0.8085	0.8092	0.8085	0.7856	
HDAWCR	0.8148	0.4066	0.4583	0.4307	

Results analysis

To ensure the scientific and rigorous design of the experiment, this study also selected four other oversampling methods: SMOTE, ADASYN and HDAWCR, as well as the mainstream machine learning models RF and GBDT, and KNN, to compare with the XGBoost model.

First, using t-SNE visualization, we compared the quantity and distribution of each class of data before and after oversampling. We found that the original dataset, after different oversampling methods, showed an increase in the amount of 2nd and 3rd grade data. It is not difficult to see that the data clustering effect after K-means-SMOTE is more pronounced and better than other oversampling methods, as shown in Fig. 6. At the same time, the effectiveness of the proposed method is demonstrated through multiple experiments, and the experimental results are shown in Table 5. Based on the data in the table, it is clear that the K-means-SMOTE-XGBoost model used in this study outperforms other models in various performance metrics. Specifically, the model achieved an accuracy of 0.8936, a precision of 0.9249, a recall of 0.8936, and an F1 score of 0.8851.

Figure 6 t-SNE visualization of (A) the original dataset and the dataset after (B) K-means-SMOTE, (C) SMOTE, and (D) ADASYN oversampling.

Based on this, and considering the feature importance ranking in the “Results” along with the biological characteristics of wheat FHB, the severity of wheat FHB each year largely depends on variations in factors such as rainfall, humidity, and temperature (Li et al., 2021, 2022; Matengu et al., 2024). Therefore, four meteorological factors were selected—AWS4.24−4.28, RAIN4.5−4.19, MTD5.14−5.18, and ARH4.12−4.16 were selected to establish a dataset incorporating these variables. Following this, the performance of different models was re-evaluated using this dataset, according to the parameter selection method outlined in the “Results”. At this point, the classification model parameters were set as follows: learning_rate = 0.03, max_depth = 3, and n_estimators = 200. The specific experimental results are shown in Table 6.

Table 6 Results of different models based on the top four ranked meteorological factors.

The bolded entries represent the best-performing results for the corresponding metrics among all the compared models.

Method	Accuracy	Precision	Recall	F1 score	
K-means-SMOTE-GBDT	0.8723	0.8894	0.8723	0.8651	
K-means-SMOTE-RF	0.8511	0.8761	0.8511	0.8395	
K-means-SMOTE-XGBoost	0.9149	0.9362	0.9149	0.9103	
K-means-SMOTE-KNN	0.8511	0.8855	0.8511	0.8210	
SMOTE-GBDT	0.8723	0.9051	0.8723	0.8545	
SMOTE-RF	0.8723	0.9053	0.8723	0.8546	
SMOTE-XGBoost	0.8511	0.8855	0.8511	0.8210	
SMOTE-KNN	0.8298	0.8602	0.8298	0.7986	
ADASYN-GBDT	0.7872	0.7810	0.7872	0.7789	
ADASYN-RF	0.7234	0.7088	0.7234	0.7119	
ADASYN-XGBoost	0.8511	0.8536	0.8511	0.8487	
ADASYN-KNN	0.7660	0.7518	0.7660	0.7512	
HDAWCR	0.8043	0.7756	0.7731	0.7712	

Finally, in order to enhance wheat FHB prevention, combined with the ranking of meteorological factors in the “Results” as well as the biological characteristics of wheat FHB, this means that wheat FHB is more likely to occur during the heading and flowering stages (Xu, 2022; Pang, 2023), in the southern region of Henan, the heading and flowering stages of wheat primarily occur from mid to late April. Therefore, this study replaced MTD5.14−5.18 with LT4.19−4.23, by concentrating meteorological factors in April, the period of high prevalence for wheat FHB. Ultimately, four key meteorological factors that contribute to the occurrence of wheat FHB were selected—AWS4.24−4.28, RAIN4.5−4.19, ARH4.12−4.16, and LT4.19−4.23 —to establish a dataset based on these four factors. Similarly, the performance of different models was re-evaluated according to the parameter selection method outlined in the “Results”. At this point, the classification model parameters were set as follows: learning_rate = 0.001, max_depth = 3, and n_estimators = 100. The specific experimental results are shown in Table 7.

Table 7 Results of different models based on the selected four key meteorological factors.

The bolded entries represent the best-performing results for the corresponding metrics among all the compared models.

Method	Accuracy	Precision	Recall	F1 score	
K-means-SMOTE-GBDT	0.8723	0.8894	0.8723	0.8651	
K-means-SMOTE-RF	0.8085	0.8548	0.8085	0.7813	
K-means-SMOTE-XGBoost	0.8936	0.9058	0.8936	0.8898	
K-means-SMOTE-KNN	0.8298	0.8684	0.8298	0.7842	
SMOTE-GBDT	0.7660	0.7851	0.7660	0.7484	
SMOTE-RF	0.8298	0.8755	0.8298	0.7866	
SMOTE-XGBoost	0.8511	0.8951	0.8511	0.8241	
SMOTE-KNN	0.8085	0.8499	0.8085	0.7644	
ADASYN-GBDT	0.7660	0.7589	0.7660	0.7484	
ADASYN-RF	0.7447	0.7255	0.7447	0.7275	
ADASYN-XGBoost	0.7447	0.7437	0.7447	0.7036	
ADASYN-KNN	0.7660	0.7785	0.7660	0.7544	
HDAWCR	0.7037	0.3489	0.3944	0.3660	

Comparing the results of Tables 5–7, it is evident that Table 6 performs better overall than Tables 5 and 7. In this case, the accuracy and recall of the K-means-SMOTE-XGBoost model in this research reached 0.9149, with precision at 0.9362 and an F1 score of 0.9103. The results in Table 6 not only validate the effectiveness of selecting meteorological factors but also demonstrate that focusing on key meteorological factors can significantly improve prediction accuracy and reliability while maintaining the simplicity of the model.

Furthermore, when concentrating on key meteorological factors during critical periods affecting wheat FHB, namely AWS4.24−4.28, RAIN4.5−4.19, ARH4.12−4.16, and LT4.19−4.23, the model shows slightly lower performance in various metrics compared to Table 6. However, compared to Table 5, the accuracy and recall remain unchanged at 0.8936, the F1 score increased from 0.8851 to 0.8898, and precision decreased from 0.9249 to 0.9058, still outperforming other models. Moreover, the selected key meteorological factors can predict wheat FHB severity levels at an earlier stage, providing a more efficient and practical model optimization strategy for accurately predicting and managing wheat FHB. For ease of comparison, line charts of predicted versus observed values and heatmaps of the confusion matrix for the K-means-SMOTE-XGBoost model before and after selecting the four key meteorological factors are provided, as shown in Figs. 7 and 8.

Figure 7 (A) shows the line graph of predicted values vs. observed values based on eight meteorological factors, while (B) shows the case based on four key meteorological factors.

Figure 8 (A) shows the confusion matrix based on eight meteorological factors, while (B) shows the confusion matrix based on four key meteorological factors.

The significance results from the t-test comparing the K-means-SMOTE-XGBoost method with other methods in terms of accuracy, precision, recall, and F1 score are presented in Table 8. Ten experiments were conducted for both the K-means-SMOTE-XGBoost model and the comparison methods under different random seeds. A P-value below 0.05 indicates a statistically significant difference between K-means-SMOTE-XGBoost and the other methods. As shown in the table, significant differences are observed across all groups for accuracy, precision, recall, and F1 score ( P < 0.05).

Table 8 Significance results from t-test comparing different methods with K-means-SMOTE-XGBoost in terms of accuracy, precision, recall, and F1 score.

Method	P-value	
	Accuracy	Precision	Recall	F1 score	
K-means-SMOTE-GBDT	4.20E–3	2.20E–3	4.20E–3	9.50E–3	
K-means-SMOTE-RF	3.80E–3	1.90E–3	3.80E–3	3.90E–3	
K-means-SMOTE-KNN	1.30E–3	2.00E–3	1.30E–3	1.00E–3	
SMOTE-GBDT	6.00E–4	8.30E–3	6.00E–4	1.20E–3	
SMOTE-RF	6.00E–4	9.00E–4	6.00E–4	3.00E–4	
SMOTE-XGBoost	4.20E–3	2.10E–3	4.20E–3	2.50E–3	
SMOTE-KNN	1.55E–2	9.00E–3	1.55E–2	7.20E–3	
ADASYN-GBDT	8.00E–4	1.42E–2	8.00E–4	1.70E–3	
ADASYN-RF	2.00E–4	6.00E–4	2.00E–4	2.00E–4	
ADASYN-XGBoost	3.10E–3	3.20E–3	3.10E–3	2.70E–3	
ADASYN-KNN	3.60E–3	1.60E–3	3.60E–3	2.40E–3	
HDAWCR	1.19E–11	1.15E–14	2.72E–15	9.63E–16	

Based on the experimental results, it can be observed that the K-means-SMOTE-XGBoost method adopted in this study demonstrates excellent performance. Specifically, this study aims to develop a new wheat FHB severity levels prediction model by integrating the K-means-SMOTE and XGBoost models. By introducing the K-means-SMOTE technique, this study effectively addresses the data imbalance issue, enhancing the model’s prediction capability for high-risk FHB levels. Meanwhile, the use of the XGBoost algorithm not only improves prediction accuracy but also optimizes the feature extraction process, enabling the model to learn more decisive information from complex climate and crop health data.

Discussion

Wheat FHB is a typical climate-dependent disease (Savary et al., 2019), and its prediction models are influenced by various factors, including cultivar resistance, meteorological conditions, and field inoculum levels. Currently, machine learning-based methods have been applied to wheat FHB prediction. Tao et al. (2021) used disease spike rate data from the Haihe Plain region, along with meteorological data during key growth stages of wheat, to perform stepwise regression analysis. This approach identified key meteorological factors influencing wheat FHB occurrence, leading to the development of an enhanced regression tree model with a prediction accuracy of 89.21%. Additionally, Shah et al. (2014) utilized the enhanced regression tree model to predict the probability of wheat FHB severity exceeding 10%, achieving a 31% reduction in misclassification rate on test data compared to a logistic regression model. However, this model failed to accurately reflect the relationship between disease spike rate and disease severity levels. In contrast, this study focuses on addressing the data imbalance issue observed in real-world production scenarios, focusing on the prediction of wheat FHB severity levels using meteorological data. The results demonstrate that the K-means-SMOTE-XGBoost model achieved the highest accuracy, precision, recall, and F1 score compared to other models, particularly after the reduction of meteorological factors to four key variables. The model achieved an accuracy of 0.8936, precision of 0.9058, recall of 0.8936, and an F1 score of 0.8898, indicating its robustness and reliability in predicting wheat FHB severity levels. This demonstrates the practical value of the K-means-SMOTE-XGBoost model for wheat FHB severity prediction.

This study’s findings are consistent with previous research that highlights the importance of addressing data imbalance and selecting key meteorological factors for improving model performance. The use of the K-means-SMOTE technique has been particularly effective in handling class imbalance, a common issue in agricultural disease prediction models. Additionally, the proposed model’s superior performance aligns with the results of similar studies that employed advanced machine learning techniques such as XGBoost for crop disease prediction. Since wheat FHB is a typical climate-dependent disease (Savary et al., 2019), it exhibits distinct regional specificity. Giroux et al. (2016) evaluated the effectiveness of nine different models in predicting wheat FHB incidence or DON toxin content in Quebec, Canada. The models developed in the United States by De Wolf, Madden & Lipps (2003), De Wolf & Isard (2007) and in Argentina by Moschini & Fortugno (1996), Moschini et al. (2001) outperformed the other models (Giroux et al., 2016), indicating that prediction model applicability may vary by region. Similarly, although the model proposed in this study has achieved good results in predicting the severity of wheat FHB in southern Henan, there are still some limitations. First, the model’s applicability is restricted due to significant climatic differences across regions. The model is primarily optimized for the climate type of southern Henan, so it may not achieve the same predictive accuracy in regions with markedly different climatic conditions, such as the colder northern regions or the wetter southern regions. Furthermore, the variability in different crops’ adaptability to meteorological conditions also limits the model’s generalizability. For example, wheat crops in the Huang-Huai-Hai region have similar climatic characteristics and growth environments to those in southern Henan (Huang et al., 2016), and the model may perform well in this area; however, its effectiveness for wheat crops in other climatic types still requires further validation.

To enhance the model’s generalizability and applicability, future research could focus on adaptability studies in regions with different climate types to adjust and optimize the model accordingly. Additionally, the model could be extended to other crops to explore the differences in response to meteorological conditions. By comparing the adaptability of different crops, the application scenarios of the model can be further enriched.

Additionally, the K-means-SMOTE-XGBoost model proposed in this study can be effectively integrated into existing agricultural monitoring systems. In Henan Province, this system collects meteorological data and crop health information in real time, using our predictive system to analyze the impact of meteorological conditions on the occurrence of wheat FHB, thereby enabling early warning. Farmers and agricultural institutions can utilize this predictive information to take timely preventive measures, reducing the impact of diseases on yield. Farmers can adjust their spraying and irrigation plans based on the predicted severity of wheat FHB, especially by taking corresponding measures in advance when the system predicts a high-risk period. Furthermore, agricultural institutions can optimize resource allocation within the region based on the predictive results, ensuring that sufficient preventive resources are available during critical periods, thereby improving overall control effectiveness.

Conclusion

This study addressed the monitoring and prediction of wheat FHB severity levels, and successfully implemented several key technologies and strategies that significantly improved the accuracy and efficiency of wheat FHB severity levels prediction. Handling of Data Imbalance: By employing the K-means-SMOTE technique, we effectively addressed the severe class imbalance issue in the dataset. This technique generated new sample points around the minority class samples, enhancing the model’s performance in predicting rare but severe wheat FHB severity levels. The experimental results show that the model’s accuracy and recall are both 0.8936, with an F1 score of 0.8898 and precision of 0.9058. This indicates that the method significantly enhances the model’s ability to identify high prevalence wheat FHB, thereby ensuring the accuracy and reliability of the prediction results. Optimization of Feature Extraction: The use of the XGBoost algorithm to analyze complex meteorological and crop health data allowed for the effective extraction of key features influencing FHB development. Through refined feature engineering and model tuning, we were able to capture the subtle impacts of meteorological information changes on wheat FHB occurrence, further enhancing the model’s interpretability and prediction accuracy. Efficient and Accurate Model Construction: By combining K-means-SMOTE and XGBoost technologies, this study constructed an efficient and accurate wheat FHB severity levels prediction model. This model exhibited not only efficient learning capabilities during the training phase but also the ability to rapidly respond and accurately predict the risk severity levels of wheat FHB in practical applications. Comparative analysis confirmed that the model maintained excellent performance under various meteorological conditions and different stages of wheat growth, effectively supporting the real-time monitoring and dynamic management of wheat FHB.

Supplemental Information

Supplemental Information 1 The main code.

kendeer.py is the code for Kendall feature selection, KMeans-SMOTE.py is the code for KMeans-SMOTE oversampling, and XGBoost.py is the code for the XGBoost classification model.

Supplemental Information 2 Data.

After Kendall feature selection, prior to oversampling. Data_KMeans-SMOTE.xlsx includes the data after KMeans-SMOTE oversampling, and Data_4Meteorological Factors.xlsx contains the data for the four key meteorological factors ultimately selected.

The authors thank the Plant Protection and Quarantine Station of Henan Province for its strong support, and thank the editor and anonymous reviewers for their helpful comments and suggestions.

Additional Information and Declarations

Competing Interests

The authors declare that they have no competing interests.

Author Contributions

Xiaoyun Sun conceived and designed the experiments, performed the experiments, analyzed the data, performed the computation work, prepared figures and/or tables, authored or reviewed drafts of the article, and approved the final draft.

Shuaiming Su performed the experiments, analyzed the data, performed the computation work, prepared figures and/or tables, authored or reviewed drafts of the article, and approved the final draft.

Qiang Wang conceived and designed the experiments, analyzed the data, prepared figures and/or tables, authored or reviewed drafts of the article, and approved the final draft.

Shufeng Xiong conceived and designed the experiments, performed the experiments, analyzed the data, authored or reviewed drafts of the article, and approved the final draft.

Yanting Li analyzed the data, performed the computation work, prepared figures and/or tables, and approved the final draft.

Hong Peng analyzed the data, authored or reviewed drafts of the article, and approved the final draft.

Lei Shi conceived and designed the experiments, performed the experiments, analyzed the data, performed the computation work, prepared figures and/or tables, authored or reviewed drafts of the article, and approved the final draft.

Data Availability

The following information was supplied regarding data availability:

The code and data are available in the Supplemental Files.

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
