# Peer review of "Prediction of wheat fusarium head blight severity levels in southern Henan based on K-means-SMOTE and XGBoost algorithms"

_PeerJ Computer Science, doi:10.7717/peerj-cs.2638_

## Round 0.1 · original submission · Minor Revisions

The three reviewers that have commented on the manuscript have expressed positive feedback. However, they make a few suggestions to further improve the quality of the paper.

Reviewer 1 ·

Basic reporting

The paper explores the use of machine learning techniques to predict Fusarium head blight (FHB) severity levels, a destructive wheat disease. It focuses on meteorological factors such as rainfall, sunshine hours, wind speed, temperature, humidity, and temperature differences, which influence FHB occurrence. The authors developed a prediction model using K-means-SMOTE to address data imbalance and XGBoost to improve prediction accuracy.

The English used in the paper is generally clear and professional. However, some complex sentence structures, particularly in the introduction and methodology sections, could be simplified to improve readability. Minor grammatical issues exist but do not significantly detract from the manuscript's quality.

The paper provides sufficient references to prior research on wheat disease prediction and relevant meteorological factors. Key studies on FHB and machine learning techniques, such as XGBoost and K-means-SMOTE, are well-cited. However, the literature review could be expanded to discuss the challenges faced in previous studies on wheat disease prediction, emphasizing the novelty of the proposed approach.

The article follows a logical structure, with well-defined sections for methodology, results, and discussion. Figures and tables are relevant, clear, and effectively support the text. However, the raw data provided is not thoroughly discussed in the manuscript. The authors should provide more details on how the raw data was processed and explain its relevance to the results more explicitly.

The paper is self-contained and presents results that directly address the research hypotheses. The proposed K-means-SMOTE-XGBoost model is tested thoroughly, and the results demonstrate that it effectively handles data imbalance while accurately predicting FHB severity levels. The discussion ties the results back to the research question effectively.

The formal results are presented clearly, with well-defined terms and metrics such as accuracy, precision, recall, and F1 score. While the study does not involve complex theorems or proofs, the methodology for K-means-SMOTE and XGBoost is explained in sufficient detail for replication. However, additional discussion on algorithmic choices and parameter settings would improve the technical rigor of the results section.

Experimental design

The paper fits well within the journal's aims and scope, particularly in its focus on applying computational methods to agriculture, machine learning, and data analysis. By introducing a combination of K-means-SMOTE and XGBoost, the research offers a significant contribution to advancing technology-driven solutions for agriculture.

The investigation is technically rigorous, utilizing appropriate machine learning techniques to address challenges posed by data imbalance and prediction accuracy. The authors provide a thorough comparison of their model’s performance against several other models, demonstrating its superiority. However, the paper does not address ethical considerations in data collection and handling, which could be important given the use of historical meteorological and agricultural data. A brief note on data ethics would strengthen the paper’s ethical foundation.

The methods are well described, making the study replicable. The steps for collecting and processing meteorological data, handling data imbalance with K-means-SMOTE, and implementing the XGBoost algorithm are clearly laid out. The specific parameter settings for the K-means and XGBoost models are also provided. However, more detail on raw data preprocessing and hyperparameter tuning could ensure full replication.

Validity of the findings

While the paper does not explicitly assess its broader impact or novelty, it implicitly demonstrates both by addressing the challenges of data imbalance in FHB prediction. The authors encourage replication by clearly stating the rationale for their research and explaining how it fills gaps in existing literature. The benefit to the field is clear, as improving prediction accuracy for underrepresented disease severity levels can significantly advance agricultural disease management.

The underlying data, including meteorological and FHB severity data, are robust, and the methods used to process the data appear statistically sound, particularly in addressing data imbalance. While the raw data is referenced, additional discussion on the robustness of data collection methods and controls to ensure accuracy and reliability would enhance the paper’s quality.

The conclusions are clear, concise, and directly tied to the research question. They summarize the main findings, particularly the effectiveness of the K-means-SMOTE and XGBoost models in predicting FHB severity with high accuracy. The conclusions reflect the experimental results and remain limited to supporting data. Additionally, the authors suggest future research directions, such as applying the model to other regions or crops, showing a thoughtful approach to extending the study’s impact without overreaching.

Additional comments

-

Reviewer 2 ·

Basic reporting

Esteemed Authors, thank you for the effort and hard work placed into the research, analysis, and manuscript preparation. While the manuscript has merit and addresses a pressing and important topic there are some concerns that should be addressed prior to the manuscript being accepted. Below are some helpful suggestions that would benefit the proposed work:

The manuscript is well-written and concise. However there seems to be an issue with spaces missing after punctuation marks such as ".", ",", ":" etc. This makes the manuscript somewhat difficult to read. Please correct this prior to resubmission. I recommend the authors proofread their manuscript and correct these and other typos.

While the literature review seems reasonable and relevant, the manuscript would definitely benefit form additional state-of-the-art works being included in the literature review. The advantages and shortcomings of recent studies in the field should also be better highlighted.

The structure of the manuscript seems a little odd. The introduction section should be separated into distinct sections. A discreet Introduction and Related Works sections should be defined. Additionally, the research gap, scientific contributions, and research question should be clearly stated.

The manuscript is fairly self-contained, with all the relevant information provided. Additionally, the supplementary materials make it easy to replicate the conducted simulations.

Experimental design

The results are well presented however, some abbreviations need to be introduced prior to their use and an experimental setup section should be present in the manuscript. Additionally, once an abbreviation is introduced it should not be introduced again in later sections (For example XGBoost in the conclusion)

While it is good that several techniques are compared in the experimental sections, the authors should justify their choice of algorithm. A comparison between mostly tree-based techniques is conducted. Could other approaches be viable?

The authors should note the computational complexities associated with each approach. Are there some advantages of using one technique over the other?

The authors should elaborate further on their choices related to the parameter pools and the solutions utilized. How were these values selected? What is the parameter sensitivity of the tested models?

The confusion matrices could benefit from better labeling. What do the 0, 1, 2, and 3 classes represent? Figures 5 and 6 are also much too small to read and interpret.

The authors should elaborate further on their choices related to the parameter pools and the solutions utilized. How were these values selected? What is the parameter sensitivity of the tested models?

The conclusion section should be expanded with the limitations of the conducted study, as well as potential future works. Additionally, performance results should be highlighted in the conclusion.

Validity of the findings

The findings seem fairly valid. However, due to the the randomness associated with the initialization procedures of the tested algorithm, if possible the authors might want to consider getting a larger sample size of outcomes, and conducting statistical validations to ensure a statistically significant improvement.

Additional comments

Esteemed Authors, thank you for the effort and hard work placed into the research, analysis, and manuscript preparation. While the manuscript has merit and addresses a pressing and important topic there are some concerns that should be addressed prior to the manuscript being accepted. Below are some helpful suggestions that would benefit the proposed work:

Reviewer 3 ·

Basic reporting

1. The introduction and methods sections, sentences tend to be long and overly complex.
2. The discussion in literature review focuses heavily on machine learning techniques. I think it could benefit from more citations to biological studies or epidemiological models on Fusarium head blight.

Experimental design

Please provide justification behind selecting specific meteorological factors (AWS, RAIN, ARH, and LT) and the use of Kmeans-SMOTE, particularly comparing it to other oversampling techniques beyond SMOTE (e.g., ADASYN).

Validity of the findings

What is the limitation of this study? Could this model be generalized to other regions outside Henan?

Additional comments

While the paper is focused on model performance, more attention should be paid to the practical application of the model. Authors should discuss how might this prediction system be integrated into existing agricultural monitoring systems in Henan? How can it be used by farmers or agricultural agencies?

---

## Round 0.2 · accepted · Accept

All reviewers are happy with this last revision, and I am therefore glad to accept this work for publication in PeerJ Computer Science.

Reviewer 1 ·

Basic reporting

no comment

Experimental design

no comment

Validity of the findings

no comment

Additional comments

The authors have addressed all my concerns.

Reviewer 2 ·

Basic reporting

The authors have addressed my concerns in the revision.

Experimental design

The authors have addressed my concerns in the revision.

Validity of the findings

The authors have addressed my concerns in the revision.

Reviewer 3 ·

Basic reporting

The authors have addressed all prior concerns regarding clarity and structure.

Experimental design

All issues related to the experimental design have been resolved.

Validity of the findings

The authors have clarified and supported their findings effectively. The results are now robust and convincingly interpreted.

Additional comments

All my previous concerns have been addressed by the authors.